# A Functional Carbohydrate Degrading Enzyme Potentially Acquired by Horizontal Gene Transfer in the Genome of the Soil Invertebrate *Folsomia candida*

**DOI:** 10.3390/genes13081402

**Published:** 2022-08-07

**Authors:** Ngoc Giang Le, Peter van Ulsen, Rob van Spanning, Abraham Brouwer, Nico M. van Straalen, Dick Roelofs

**Affiliations:** 1Department of Ecological Science, Faculty of Science, Vrije Universiteit, 1081 HV Amsterdam, The Netherlands; 2Department of Molecular Cell Biology, Faculty of Science, Vrije Universiteit, 1081 HV Amsterdam, The Netherlands; 3BioDetection Systems, 1098 XH Amsterdam, The Netherlands; 4Keygene N.V., 6708 PW Wageningen, The Netherlands

**Keywords:** soil invertebrates, Collembola, *Folsomia candida*, horizontal gene transfer, microbiome, metagenomics, CAZy, arabinofuranosidase, signal peptide, co-evolution

## Abstract

Horizontal gene transfer (HGT) is defined as the acquisition by an organism of hereditary material from a phylogenetically unrelated organism. This process is mostly observed among bacteria and archaea, and considered less likely between microbes and multicellular eukaryotes. However, recent studies provide compelling evidence of the evolutionary importance of HGT in eukaryotes, driving functional innovation. Here, we study an HGT event in *Folsomia candida* (Collembola, Hexapoda) of a carbohydrate-active enzyme homologous to glycosyl hydrase group 43 (GH43). The gene encodes an N-terminal signal peptide, targeting the product for excretion, which suggests that it contributes to the diversity of digestive capacities of the detritivore host. The predicted α-L-arabinofuranosidase shows high similarity to genes in two other Collembola, an insect and a tardigrade. The gene was cloned and expressed in *Escherichia coli* using a cell-free protein expression system. The expressed protein showed activity against p-nitrophenyl-α-L-arabinofuranoside. Our work provides evidence for functional activity of an HGT gene in a soil-living detritivore, most likely from a bacterial donor, with genuine eukaryotic properties, such as a signal peptide. Co-evolution of metazoan GH43 genes with the Panarthropoda phylogeny suggests the HGT event took place early in the evolution of this ecdysozoan lineage.

## 1. Introduction

In general, a genome is passed from parents to offspring, and its DNA sequence reflects the evolutionary history of the lineage. However, genomes are dynamic and can be altered through loss or gain of genes or expansion or contraction of non-coding or transposable elements. Genes can be gained through duplication or acquired from foreign sources by horizontal gene transfer [1]. Horizontal gene transfer (HGT) is a mechanism by which organisms may acquire functions that can hardly be obtained by selection on standing genetic variation. The frequency of successful HGT depends on the ability of the host to take up the foreign DNA from its environment, the ease at which the foreign DNA can recombine and integrate with host genomic DNA, and the access of foreign donor DNA to the germline of the host [2].

Once integrated, the newly acquired DNA is subjected to evolutionary pressure and selection. Only DNA that provides new functionalities to the host or contributes to existing functions will be maintained over generations. In addition, such genes are often adapted to the host genome. In contrast, non-beneficial DNA acquisitions are lost over time. Especially for bacteria, HGT is an important mechanism for evolutionary innovation and the exploitation of new habitats [2,3].

There are several ways by which genes can be transferred from one organism to another: through transformation, transduction, or bacterial conjugation, or by gene transfer agents [3]. Conjugation implies that donor and recipient are in physical contact and genetic material is exchanged through a conjugation pilus. For instance, *Agrobacterium* spp. uses conjugation to transfer T-DNA to plant cells [4]. Transformation implies that environmental DNA is taken up by the recipient. Transduction is a mechanism in which phages or viruses deliver genetic material to the recipient. All of these mechanisms are observed in archaea and bacteria and have been crucial for the evolution of both phyla [5].

While the evolutionary consequences of HGT in bacteria are significant, HGT is extremely rare among eukaryotes, and uncommon between prokaryotes and eukaryotes [6]. Studies on newly derived genome sequences of non-model animals have claimed examples of HGT from bacteria into eukaryotes [2]. Several solid cases for HGT in invertebrates have been made, including nematodes, tardigrades, rotifers, and springtails [7,8,9]. Most recent studies have been conducted by using long read single molecule sequencing data, which provide physical linkage of the donor gene in an eukaryotic genome [10]. Additional phylogenetic analysis can provide further evidence to ascertain HGT cases [11].

The hexapod class Collembola (springtails) has been shown to be a hot spot of horizontal gene transfer [10]. In the genome of the model species *F. candida,* the percentage of open reading frames due to horizontal gene transfer was estimated as 2.8% after thorough validation [10]. Since springtails live in close proximity with soil microbial communities and because they evolved as an ancestral group of hexapods, the opportunity for HGT is realistic [3]. The class Collembola includes several species capable of anhydrobiosis, a mechanism of extreme drought-tolerance that includes dissolution of the nuclear membrane and partial fragmentation of the genome. Anhydrobiosis has been suggested as a mechanism of HGT in nematodes, tardigrades, and bdelloid rotifers [2].

A recurrent question in HGT cases is how the (usually prokaryotic) donor DNA can not only be inserted but also expressed in a eukaryotic genomic environment. Only in a few cases has it been demonstrated that sequences of prokaryotic origin are actually expressed in the eukaryotic host genome [12]. These events have occurred in the evolutionary past and most likely continue to occur to shape eukaryotic genomes. Among the various genes acquired by HGT in springtails, biosynthesis clusters for β-lactam antibiotics are one of the most striking [13]. However, another important functional contribution of HGT is due to carbohydrate-active enzymes. Carbohydrates are needed for multiple biological purposes, such as energy storage, signal transduction, and intracellular trafficking [14]. They are also the future of renewable energy, such as biofuel production [15]. Identification of potentially novel enzymes that can break down biomass are therefore of prime interest. Obviously, the degradation of recalcitrant biomass is an extremely important capacity for any detritivore soil invertebrate [16]. They may therefore represent a valuable genetic resource to discover novel enzymes that can convert biomass into compounds amenable for bio-renewable energy production.

The carbohydrate-active enzymes (CAZymes) form a diverse group of proteins that function in carbohydrate metabolism. The carbohydrate-active enzymes database (CAZy) is the largest and most well-annotated sequence-based classification system [17]. It includes glycosyl hydrolases (GHs), glycosyltransferases (GTs), polysaccharide lyases (PLs), carbohydrate esterases (CEs), carbohydrate-binding modules (CBMs), and auxiliary activities (AAs). These proteins jointly are responsible for the breakdown of lignocellulose, an abundant carbohydrate resource in soil ecosystems. We previously applied this database to identify carbohydrate-active enzymes in the metagenome of springtails and their associated microbiome [9,10]. Also, we were able to link several CAZy-encoding genes in the host genome to a putative microbial donor and thus to a putative HGT event. We also showed that most of them are transcribed and therefore active.

Here, we further characterize one of these putative HGT-derived CAZymes: α-L-arabinofuranosidase. This enzyme catalyzes the cleavage of L-arabinose side-chains in hemicellulose, an important step in the final breakdown of this poly-sugar compound into monomeric sugars. In this paper, we provide functional evidence of its activity in a cell-free environment. In addition, we show that the gene shows adaptive evolution most likely to facilitate extracellular activity of the enzyme in the gut lumen of the collembolan host.

## 2. Materials and Methods

### 2.1. Gene Annotation

Previously analyzed and cataloged transcripts from *F. candida* were downloaded from the Collembolomics genome web browser [10] for CAZy annotation analysis [10]. Prodigal was applied on transcript sequences to predict bacterial open reading frames (ORF). Proteins with start and stop codons were scanned against the CAZy database using the Hidden Markov Model (HMM) algorithm with default settings [18]. The CAZyme candidate genes were further explored using the basic local alignment search tool (BLAST) software from the National Center for Biotechnology Information (NCBI) to establish sequence homologies [19]. The binding site structures were predicted using the Phyre2 web server [20] along with the SWISS-MODEL package [21] using default settings. Candidate genes were selected based on high homology of the bacterial CAZy family group and binding sites. We focus here on an *F. candida’s* open reading frame showing high similarity to α-L-arabinofuranosidase from *Lactobacillus lactis* (Appendix A). Nucleotide sequences were analyzed for the presence of signal peptides using Gram negative and Gram positive settings with the SignalP4.1 server [22]. Protein molecular weight and isoelectric point (pI) value calculations were performed using Cloning Manager 9.0 (Sci-Ed Software, Westminster, CO, USA). The protein was blasted against the springtail proteins and transcript data at https://collembolomics.nl/ (accessed on 29 December 2020) using default settings [10].

### 2.2. Plasmid Construction for Recombinant Expression

Total RNA from whole springtails was extracted using the SV Total RNA isolation system according to the manufacturer’s protocol (Promega, Madison, WI, USA). Subsequently, mRNA was converted to cDNA using oligo dT(15)-guided reverse transcription with AMV reverse transcriptase according to manufacturer’s instructions (Promega, Madison, WI, USA). PCR was performed on cDNA by applying the following oligonucleotide primers designed on the predicted α-L-arabinofuranosidase (FcAraf43) gene from the ORF of *Folsomia* transcript: 5′-primer (5′-GGGCATATGGCTTTCACAAAAATATTG-3′), which included the ATG translational start codon inside a NdeI restriction site (shown in italic) and 20 nucleotides of the ORF. The 3′-primer (5′-AAACTCGAGTTATTCCCCACTTGGAAC-3′) included a stop codon (TAG), containing an XhoI restriction site and the preceding 26 nucleotides of the ORF. Three guanine and thymine residues were added at the 5′-end of the 5′-primer and 3′-primer, respectively, to create a good binding site for the respective restriction enzymes. The gene sequence was amplified using Taq and Pfu polymerases, and the product was purified on a 1% agarose gel. The product was digested with NdeI and XhoI and ligated into NdeI/XhoI-digested pET16b vector, resulting in the plasmid pET16-FcAraf43 with an N-terminal His-tag. The resulting plasmid was transformed into XL1-blue chemically competent cells. Successfully transformed colonies were screened by restriction digestion and confirmed by DNA sequencing (Macrogen, Seoul, Korea) before being transformed in *E. coli* expression strain Rosetta2 (DE3) (Novagen, Madison, WI, USA).

The ORF encoding the processed form of FcAraf43 was obtained by PCR on pET16-FcAraf43 plasmid as template DNA by applying primers (5′-TAATACGACTCACTATAGGG-3′) and (5′-GCTAGTTATTGCTCAGCGG-3′). The PCR amplicon was subsequently cloned into the pGEM-T vector (Promega) following the instructions of the manufacturer. Successful amplification and cloning were confirmed by sequencing. The ORF was then further subcloned into pET16b using the NcoI and BamHI restriction sites designed in the primer sequences.

### 2.3. Recombinant Protein Expression and Purification of FcAraf43

A glycerol stock of transformed Rosetta2 cells was used to inoculate 200 mL of LB medium and 100 µg/mL of ampicillin at 37 °C. Cells were cultured until the optical density at 600 nm reached 0.6–0.8. The cultures were induced by adding 50 µM of isopropyl-β-D-thiogalactopyranoside (IPTG) for gene expression and further incubated for 2 h at 37 °C. After centrifugation, the cells were harvested and suspended in 8 mL of phosphate buffer saline (PBS; pH 7.4). Protease inhibitors cOmplete™, EDTA free (Roche, Mannheim, Germany) cocktail, were added followed by two passages at ~1.7 k psi through a OneShot cell disruptor (Constant Systems Ltd., Northants, UK) at room temperature. The debris and membrane fragments were removed from the cell extract after centrifugation at 586× *g* for 10 min and 100,000× *g* for 1 h, respectively. TALON Superflow resin (GE, Uppsala, Sweden) premixed with buffer A (50 mM of potassium phosphate buffer, 500 mM of sodium chloride, 10% glycerol, 10 mM of imidazole pH 7.5) was added to the cleared cell extract and mixed. The mixture was incubated at 4 °C, for 1 h and transferred to a disposable 5-mL polypropylene column (Thermo Scientific, Waltham, MA, USA) to be washed with 10 mL of buffer A. Several wash solutions with increasing imidazole concentration up to 200 mM were used. The His-tagged proteins were eluted from the beads by adding 10 mL of buffer B (50 mM sodium phosphate buffer, 500 mM sodium chloride, 10% glycerol, 400 mM imidazole pH 7.5). To concentrate the sample and remove salts, a Vivaspin 20, MWCO 10 kDa column was used. About 20 mL of PBS at pH7.4 was added and centrifuged at 6000× *g* for five times, after which the retentate was collected and aliquoted. The BCA protein assay kit (Thermo Scientific, Waltham, MA, USA) with bovine serum albumin (BSA) as the standard was used to measure the concentration of purified protein. For displaying protein, the crude extracts or purified protein samples were denatured in sample buffer with dithiothreitol (DTT), boiled for 10 min and applied to 10% gradient sodium dodecyl sulfate polyacrylamide gel electrophoresis (SDS-PAGE, BIORAD, Hercules, CA, USA) along with the molecular weight marker to determine the molecular weight and purity. The gel was stained with 0.1% Coomassie blue as previously described by Lämmli [23].

### 2.4. Cell-Free Protein Expression

Plasmids were extracted using the Genejet plasmid purification kit (Thermo Scientific) and subjected to the PURE protein expression system (New England Biolabs, Ipswich, MA, USA) according to the manufacturer’s protocol. As a positive control, an active α-L-arabinofuranosidase gene from *Lactococcus lactis* was also cloned into pET16 (Merck, Kenilworthm, NJ, USA). The mixture was incubated at 37 °C overnight. The proteins were washed twice in PBS at pH 7.4 using spin columns to concentrate. A sample was mixed with 2× sample buffer and analyzed on SDS-PAGE. The remainder was used for enzyme-activity testing.

### 2.5. Enzyme Assays for FcAraf43

Synthetic p-nitrophenyl-α-L-arabinofuranoside (pNP-α-L-Araf) was purchased from Megazyme International (Wicklow, Ireland). α-L-arabinofuranosidases catalyze the release of p-nitrophenol (pNP) from pNP-α-L-Araf, which can be detected at 405 nm. Each assay mixture contained 10 µL of a 25 mM pNP-α-L-Araf solution with 88 µL of PBS buffer (pH 7.4) and 2 µL of enzyme solution. The reaction was carried out at 37 °C, while monitoring pNP accumulation overnight in a Synergy HTX plate reader (BioTek, Winooski, VT, USA). Read-outs in blanks were subtracted from sample reads. The assay activity was performed in triplicate unless otherwise stated.

### 2.6. Phylogenetic Analysis

Sequences characterized as GH43 enzymes from the CAZy database were used to create a phylogenetic tree, and accession numbers are included in the taxon names of the tree. Moreover, we included GH43 peptide sequences that represent the top 50 BlastP hits when using FcAraf43 as query to search the non-redundant protein database from NCBI. Phylogenetic analysis was performed by MEGA (version 7.0, Sudhir Kumar and Koichiro Tamura, Philadelphia, PA, USA) software applying maximum likelyhood (ML) and neighbor joining (NJ) analyses [24]. First, the MUSCLE algorithm within MEGA7 was used to align the input protein sequences. In case of ML, the most optimal evolutionary model for protein evolution for the input protein alignment was estimated with the model testing module within the MEGA7 package. This turned out to be the Whelan And Goldman (WAG) model [25] with γ distribution. Subsequently, both algorithms were run for 1000 bootstrap replicates [26] to obtain confidence in the branching pattern. The GH43 accession no. SCF26596.1 *Micromonospora echinospora* was taken as outgroup, as it was most divergent from all GH43 peptides in the alignment, based on amino acid differences.

## 3. Results

The ORF encoding the FcAraf43 protein of 343 amino acids was predicted from the Fcan01_09776-PA transcript [10] (NCBI assession no. XP_021951599.1). Analysis using the Collembolomics genome web browser (https://collembolomics.nl/, accessed on 29 December 2020) showed that the FcAraf43 open reading frame of 1029 bp length mapped back to scaffold 4 (Fcan01_Sc004) of the *F. candida* genome sequence with 100% identity. Further analysis indicated a predicted polyadenylation signal (AATAAA) 79 bp downstream of FcAraf43 ORF, suggesting that the gene underwent eukaryotization after the transfer from a prokaryotic donor.

The core of the protein sequence matches glycosyl hydrolase group 43 group 1 in the CAZy database (Figure 1, Appendix A). Alignment of FcAraf43 against the non-redundant protein database using blastP shows that it is closely related to gene sequences in the sister species *Orchesella cincta* (ODM95222.1), the springtail *Allacma fusca* (CAG7834605.1), the midge *Bradysia coprophila* (XP_037044875.1), and the bacterium *Thermoanaerobacterium thermosaccharolyticum* (WP_015311875.1), with 68.31%, 68,82%, 51.69%, and 40.17%, respectively. The FcAraf43 sequence is also 78% identical to a protein predicted from the Fcan01_18043-PA transcript and both are annotated as an α-L-arabinofuranosidase from *Streptomyces chartreusis* (https://collembolomics.nl/, accessed on 29 December 2020). In the *F. candida* genome, FCAraf is surrounded by annotated gene sequences that show high homology to animal-derived genes, such as 3-phosphoinositide-dependent protein kinase 1 (*Caenorhabditis elegans*), pyrimidodiazepine synthase (*Drosophila melanogaster*), and carboxypeptidase N catalytic chain (*Rattus norvegicus*). Furthermore, FcAraf43 is physically connected to these animal-derived genes by PacBio single molecule long read sequences at on average 70× read coverage. Genomic scaffold Fcan01_Sc004 (12.87 Mbp) is genuinely eukaryotic and does not represent bacterial contamination. Finally, the GC content (37.5%) of the scaffold is also according to what is expected in a eukaryotic multicellular organism.

An N-terminal signal peptide of 19 amino acids was predicted to be present in the encoded FcAraf43 protein (Figure 1). This strongly suggested that the protein is targeted for excretion to the extracellular matrix. The three-dimensional structure predicted by the Phyre2 algorithm showed a five bladed-β propeller as also found in the GH43 group of enzymes. Furthermore, two of the three predicted active site residues, corresponding to amino acid positions 45 and 154, are aspartic acid residues while the third, at position 215, is a glutamic acid. These positions are conserved within this class of enzymes, including the α-L-arabinofuranosidases. These three conserved amino acid residues act as general acid and pKa modulators [27]. Together, they ensure the inverting glycoside hydrolase reaction, which is characteristic for GH43 CAZymes. These conserved regions, including the five-bladed β-propeller fold, were structurally and functionally elucidated in L-arabinanase 43A of *Cellvibrio japonicas* [27].

Alignment analysis with homologous proteins from bacteria and fungi present and annotated in the CAZy database shows that both aspartates are fully conserved. The glutamate position, however, differs between prokaryote and eukaryote microbial versions of the enzyme (Appendix A). As observed in the protein sequence tree (Figure 2), FcAraf43 clusters with two collembolan arabinofuranosidases of *Orchesella cincta* and *Allacma fusca*, along with the midge and tardigrade GH43 peptides in a monophyletic group with bootstrap support of 91% in the ML analysis [28]. The clade is also supported when using the NJ algorithm [29], with a similar bootstrap support of 92%. The GH43 peptides most closely related to animal-derived GH43 peptides consist of two bacterial arabinofuranosidases from *Thermoanaerobacterium* and *Paenibacillus* with bootstrap support of 78% (98% support in case of NJ analysis). This suggests that the origin of animal GH43 evolution could be a single HGT event from a bacterial donor into the Panarthropoda lineage before the divergence of Arthropoda, Tardigrada, and Onychophora. Consequently, a long evolutionary history after the HGT event is suggested in *F. candida*, which may have been accompanied by adaptive evolution within the animal’s genome (Figure 2).

To allow for functional analysis, the gene encoding FcAraf43 was cloned into the pET16b expression vector and expressed in *E. coli* expression strain BL21. However, this yielded a protein band at ~35kDa on Coomassie-stained SDS-PAGE and not a band at the expected size of 42kDa for the complete ORF and its N-terminal His-tag extension that resulted from the cloning strategy chosen (Figure 3A, lane 3). In theory, this lower position could be due to processing of the predicted signal peptide, but that would have yielded a protein product of about 39 kDa. Apparently, the expression of the ORF resulted in degradation of the protein. In contrast, expression of protein LcAraf43 from a similar plasmid yielded a protein band of ~39 kDa, which is only slightly lower than the calculated molecular height of 40kDa. Note that the *L. lactis* enzyme has no signal peptide and is expected to be cytoplasmic (data submitted). We are uncertain about the nature of the ~35kDa band detected in the FcAraf43 expressing bacteria. It could be a degradation product of FcAraf43, but also could be an upregulated bacterial protein, like for example an elongation factor [30].

We then decided to clone the sequence encoding the mature part of the FcAraf43 protein (mAraf43), consisting of the domains downstream of the predicted signal sequence cleavage site. This allowed enzymatic activity of the protein to be studied in the form that is anticipated to be active. Furthermore, we decided to express this mFcAraf43 construct using the cell-free PURE expression system to avoid degradation. As a positive control, we included the plasmid encoding the LcAraf43 protein from *L. lactis*. Expression using the PURE system resulted in clearly detectable protein bands on Coomassie-stained SDS-PAGE gels. For both LcAraF43 and FcAraf43, it yielded a product of ~39kDa, which complies with the calculated molecular weight of the proteins encoded on the plasmids used. This results also implied that expression of the LcAraf43 control in Rosetta also resulted in some clipping of the protein. 

The proteins produced in the PURE system were then tested for enzymatic activity by measuring the turnover into the pNP product of the pNP-α-L-Araf substrate (Figure 3B). The LcAraf43 control, with known enzymatic activity toward the substrate clearly showed an increase in pNP when compared to the empty vector control. In support of the assignment of FcAraf43, we also see an increase when the mFcAraf43 is incubated with the substrate, although to a lesser extent. Nevertheless, the clear increase of product and the low amount of enzyme added strongly suggest that indeed, FcAraf43 shows furanidase activity.

About two µL of enzyme solution was added to the substrate to bring the total volume to 25 µL (as per instruction by the Megazime protocol). Still, this small amount of protein showed activity even though it was not as high as in the positive control from *L. lactis*.

Further analysis of the protein was performed through gel electrophoresis (Figure 3A). Additional mass spectrometry analysis shows that the start and end peptide of the protein were intact. The bands appearing at 37 kDa were identified as elongation factor [30].

The absorbance of the FcAraf43 at 405 mm is 0.203 (Figure 3B). This shows that FcAraf43 is active (center bar, Figure 3B), but at a lower rate than the *Lactococcus* positive control (right bar), which was measured at 0.280. We speculate that this could be due to the eukaryotic nature of the gene, which may hamper optimal expression in a bacterial host background (*E. coli)*.

## 4. Discussion

We have identified a functionally novel gene in *F. candida* that encodes an enzyme with α-arabinofuranosidase activity. This gene potentially entered the host by horizontal gene transfer and shows clear evidence of eukaryotization (poly-adenylation recognition sequence and an N-terminal signal peptide).

The gene was identified through the analysis of the active transcriptome of the springtail [10], and sequence analysis predicted its encoded enzyme belongs to the glycosyl hydrolase group 43. We therefore named the encoded enzyme FcAraf43. The predicted 3D structure from FcAraf43 shows a 5-bladed β-propeller structure that is common to CAZymes in this group. The enzyme contains two aspartic acid and two glutamic acid residues, which are proposed to constitute the active site [27,31,32], which shows homology to the fungal subgroup of these enzymes when comparing the variable binding mechanisms and amino acids in the catalytic center. The enzyme α-L-arabinofuranosidase is used in many industries such as food, animal feed and wine [15,33,34]. By identifying a new variant of this protein, it is possible to extend our knowledge of this group of enzymes and its evolution as well as a possible industrial application in the breakdown of lignocellulose-containing biomass.

The gene shows multiple characteristics to be the result of an HGT event. With only 1029 bp it is a relatively small gene; small inserted DNA fragments are often better tolerated in a host genome [2]. The GC content of FcAraf43 is 47%, which is higher than the average value of 37.5% in the springtail nuclear genome [10]. FcAraf43 is predicted to be an old HGT gene as it is positioned at the tip within the monophyletic group splitting off from a potentially bacterial ancestor (Figure 2). Moreover, it is located on scaffold 4 of the *F. candida* genome assembly [10], which is a gene-rich region with the most abundant density of HGT genes. This scaffold is also rich in DNA transposon and retrotransposon sequences, potentially facilitating HGT [2].

Phylogenetic analysis showed that FcAraf43 clusters within the animal GH43 lineages, which forms a monophyletic clade within bacterial sequences, while the fungal sequences cluster separately. The clustering of FcAraf43 with two bacterial arabinofuranosidases of *Thermoanaerobacterium* and *Paenibacillus* is supported by phylogenetic analysis of both ML and NJ algorithms with high confidence, suggesting a bacterial origin from a single HGT event. Within the animal clade, we observe tardigrades to be a sister group of midge- and collembolan GH43 peptide sequences. This suggests that the GH43 protein sequence tree presented here follows the common scheme for the evolution of Panarthropoda within the Ecdysozoa clade, whereby tardigrades are ancestral to arthropods [35]. The origin of FcAraf is more clear when compared to isopenicillin N synthase, another HGT gene in the *F. candida* genome [13]. This gene could not be assigned to a bacterial donor nor to a fungal donor. This supports the notion as pointed out by McDonald and Currie 2017, that HGT events are quite rare, and the location of the source of the event may depend on different evolutionary rates [36]. Finally, the signal peptide, as elucidated from Araf43, is not present in the bacterial GH43 peptides, but has been identified in fungal *Chrysosporium* and *Penicillium* [37,38]. This suggests that the recruitment of a signal peptide for GH43 happened at least two times independently in fungal and animal genomes.

One of the reasons why the gene was maintained in the host genome may be due to its advantage to host nutrition as well as exploring the host into new niches, such as in helping digesting hemicellulose [39]. Biomass is an abundant source of energy. Effective degradation of this energy source will be very beneficial to the host. Having this gene helps the springtail to breakdown hemicellulose and decrease its dependence on the gut microbiome. There are other cases where HGTs gene have been found in relation to herbivorous insect. Although these events are very rare, they do occur. A functional mannanase was found in the coffee berry borer beetle, *Hypothenemus hampei*. This gives the beetle advantages in degrading the polysaccharide of the coffee seeds [12]. Endoglucanases and pectinases were also observed in plant parasitic nematodes [40]. Another common feature found in these cell wall or other glycan degrading enzymes is that they are secreted. This was observed in other hosts, such as nematode [41], spider mite [42], and parasitic wasp [43]. The signal peptide found was the Sec/SPI secretory signal peptide. It is transported by the Sec translocon and cleaved by signal peptidase I (Lep) [22].

## 5. Conclusions

We have identified an active novel HGT-derived α-L-arabinofuranosidase in the genome of the springtail *F. candida*. The HGT gene may help this soil-living animal to digest hemicellulose from plant biomass. This in turn helps the springtail to utilize recalcitrant polysaccharides and help it to survive and thrive in the soil environment, which is known for its resource-limited characteristics.

## Figures and Tables

**Figure 1 genes-13-01402-f001:**
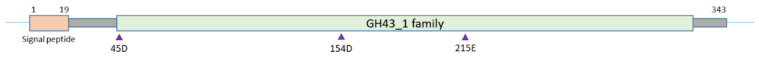
Schematic representation of the domain structure of FcAraf43. A predicted signal peptide in the first 19 amino acids is represented by a pink colored box. The green colored box represents the conserved region characteristic for the CAZy GH43 family 1 (α-L–arabinofuranosidase enzymes). The purple triangles indicate the locations of residues that constitute the active catalytic site of the enzyme.

**Figure 2 genes-13-01402-f002:**
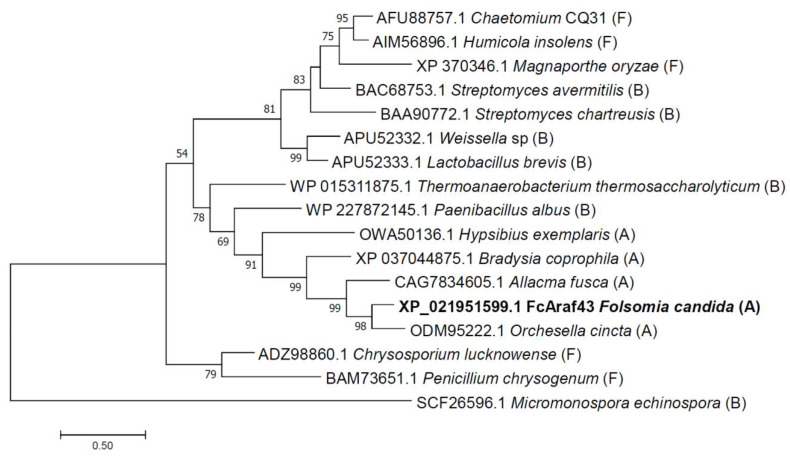
Phylogenetic relation of FcAraf43. An elaborate set of 16 animal, fungal and bacterial peptides from classified CAZy family 43 enzymes were aligned along with FcAraf43. The presented phylogeny is based on maximum likelihood using WAG+G evolutionary model; the outgroup is set at *Micromonospora*. The numbers next to branches show the percentage bootstrap support based on 1000 replicates. GenBank accession numbers of the annotated CAZy genes are given in front of species names. Branch lengths are measured as the number of substitutions per position/site, with the scale bar representing the length for 0.5 substitutions. Letters in brackets indicate phylum: A, Animal; B, Bacterial, F, Fungal.

**Figure 3 genes-13-01402-f003:**
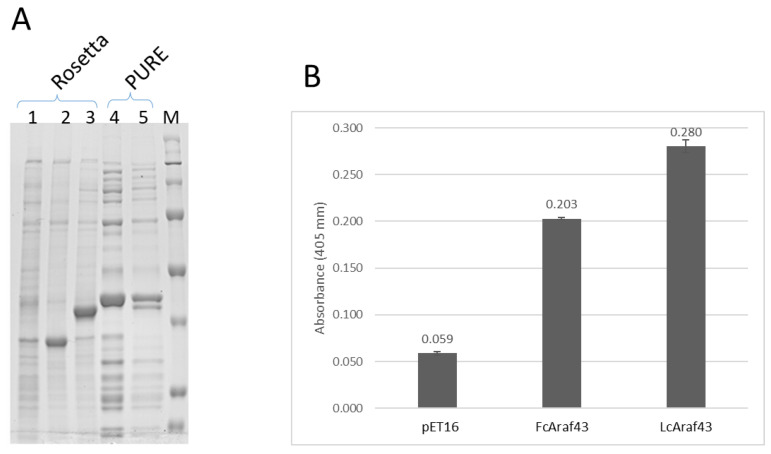
Expression and enzyme activity of FcAraf43. (**A**) Coomassie-stained SDS-PAGE gel of expression of the processed form of mFcAraf43 and the ***L. lactis*-**derived LcAraf43 control with known furanosidase activity expressed in the cell-free PURE system (lanes 4 and 5) and the full-length form of pET16b, FcAraf43 and LcAraf43 (lanes 1–3) in Rosetta (DE3). Lane 1: Negative control empty pET16 vector, Lane 2: Cell culture of FcAraf43 in BL21, Lane 3: Positive control cell culture of BL21-LcAraf43, Lane 4: FcAraf43 in PURE system, Lane 5: LcAraf43 expressed using PURE cell-free system and molecular weight standard. (**B**) Enzymatic activity of the proteins expressed in the PURE systems, including an empty vector control. The samples were incubated at 37 °C for 16 h and then measured at 405 nm to detect the amount of pNP. The bars represent the mean of three replicates, and the standard deviation is indicated by the error bars.

## Data Availability

Not applicable.

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
