# Peer review of "A Functional Carbohydrate Degrading Enzyme Potentially Acquired by Horizontal Gene Transfer in the Genome of the Soil Invertebrate Folsomia candida"

_genes, 2022, doi:10.3390/genes13081402_

Round 1

Reviewer 1 Report

I liked the revised work of Le at al. a lot more! I think authors have adequately addressed most of my concerns. Now I have some (mainly) cosmetic suggestions.

1) Ln. 22. cell-free.

2) Lns. 45-52. Please provide references.

3) Ln. 199. Mention the abbreviation WAG. 

4) Ln. 281. I am not I understand (data submitted). 

5) Fig. 1B. Please remove bootstrap values below 75. They are not very informative.   

Author Response

1) Ln. 22. cell-free.

Response: done, see line 23.

2) Lns. 45-52. Please provide references.

Response: three references were included to support our statements in this section, lines 50-57.

3) Ln. 199. Mention the abbreviation WAG.

 Response: done, see line 208.

4) Ln. 281. I am not I understand (data submitted). 

Response: we did not understand this comment.

5) Fig. 1B. Please remove bootstrap values below 75. They are not very informative. 

Response: we took out bootstrap values below 50, because it is common practice. Besides that, the tree is very well resolved and the majority of branches are supported by high bootstrap values as can be observed in figure 2. 

Reviewer 2 Report

The manuscript identified an active novel HGT alpha-L-arabinofuranosidase from the collembolan Folsomia candidaThe manuscript is well written, and I advocate its publication after minor revision.

In 2.1 Gene annotation, potential bacterial genes were predicted against all transcript sequences of Folsomia candida. Its better to show all genes annotated in this step in the Result, and then tell us why did you choose FcAraf43 for further study.

For the phylogenetic analysis, why did you use SCF26596.1 as the outgroup?The authors emphasized in the Abstract and Results that FcAraf43 has high similarity to genes in the collembolan Orchesella cincta, the midge Bradysia coprophila, the fungus Allacma fusca, and the bacterium Thermoanaerobacterium thermosaccharolyticum. Why not include these sequences in the phylogenetic analysis? It’s better to add more representative taxa from bacteria, fungi and animals to show the molecular evolution.

Fig.2 is too small to see clearly.

Line 139, The ORF encoding the processed form of FcAraf63 was obtained by PCR pET16- FcAraf43 as the template by using primers.” FcAraf63 should be FcAraf43? 

Author Response

In 2.1 Gene annotation, potential bacterial genes were predicted against all transcript sequences of Folsomia candida. It’s better to show all genes annotated in this step in the Result, and then tell us why did you choose FcAraf43 for further study.

Response: we included a supplement file 1 to show the functional basis on which Araf43 was chosen. This was added in the text in lines 119-121.

For the phylogenetic analysis, why did you use SCF26596.1 as the outgroup?

Response: SCF26596.1 was the most divergent Araf43 among the 17 Araf43 peptides included in the phylogenetic analysis based on its amino acid polymorphism. This information was added to Materials and Methods lines 210-212.

The authors emphasized in the Abstract and Results that FcAraf43 has high similarity to genes in the collembolan Orchesella cincta, the midge Bradysia coprophila, the fungus Allacma fusca, and the bacterium Thermoanaerobacterium thermosaccharolyticum. Why not include these sequences in the phylogenetic analysis? It’s better to add more representative taxa from bacteria, fungi and animals to show the molecular evolution.

Response: we included the suggested species as well as representative taxa from the top 50 BlastP hits using FcAraf43 as query. This resulted in a new phylogenetic tree consisting of 17 species, of which the phylogenetic relationships shifted when compared with the phylogeny in the previous manuscript version. We therefore adapted Results and Discussion according to the new evolutionary insights. Please, see yellow highlighted sections in Results and Discussion.

Fig.2 is too small to see clearly.

Response: we increased the size of the tree of figure 2 and moved the alignment figure to supplement 2, so that the phylogeny receives more attention.

Line 139, “The ORF encoding the processed form of FcAraf63 was obtained by PCR pET16- FcAraf43 as the template by using primers.” FcAraf63 should be FcAraf43? 

Response: this should indeed be FcAraf43. We changed this accordingly (line 147)

This manuscript is a resubmission of an earlier submission. The following is a list of the peer review reports and author responses from that submission.

Round 1

Reviewer 1 Report

The presented work of Le at al analyzed an interesting case of potential HGT of a-L-arabinofuranosidase into genome of F. candida. Overall, the work has some merit but it lacks numerous important controls and (in this reviewer's opinion) the analysis must be repeated "from scratch". In addition, many paragraphs must be rephrased to convey the intended meaning (f.e. lns 36-41, 5-61, etc. etc.) 

Major points:

A) Figure 1 is redundant and must be deleted or moved to the supplements. 

B) Phylogenetic analysis must be redone from scratch using adequate methods (no fast bootstraps, please). How these sequences were selected? This must be properly explained. 

C) Figure 3 makes very little sense. What's going on in E. coli? This must be investigated more. I suggest trying different vectors and presenting data for no IPTG stimulation. By the way, calling it in vivo is not correct. 

D) Figure 4 does not tell much about enzymatic activity. I suggest purifying the enzyme and presenting proper enzymological data. 

Minor points. 
A) English!! 

B) Make sure species and genera names are Italicized and spelled correctly. 

C) Methods describe many very trivial details but omit some important ones (f.e. recloning into pGEMT vector) 

D) Why a band around 37 kD is a housekeeping gene from E. coli? This is not true, or it must be present in all lanes 4-6.    

Reviewer 2 Report

The authors detected one horizontally transferred gene FcAraf43 in the springtail Folsomia candida and performed protein expression and enzyme assays to confirm its function. Overall, the finding is interesting. However, the writing needs to be improved and the experiment design and analysis didn't provide strong evidence to confirm the HGT event.

comments:

  1. The authors localized the FcAraf43 gene on a genome scaffold of Folsomia candida. From the text, it is not clear that the scaffold is eukaryotic or bacterial contamination. The authors need to describe if FcAraf43 is flanked by eukaryotic genes.
  2. In the phylogenetic analysis method,  it is not clear what taxa were used to perform the analysis and what out group was used. The out group was not chosen by the program (correct it on Line 280).
  3. A large number of references are missing in the introduction.
  4. In the abstract, change 'and event' to 'an event'.
  5. In the abstract, change 'we shows' to 'we show'.
  6. Line 45, if you want to use Agrobacterium app-plant as an example, you need to remove the sentence 'This process.....bacteria'.
  7. Line 230, 'Streptomyces chartreus' needs to be italic.
  8. Line 281, add 'from' before 'Streptomyces chartreus'.